# The skeletal muscle circadian clock regulates titin splicing through RBM20

**Lance A Riley[1,2†], Xiping Zhang[1,2], Collin M Douglas[1,2], Joseph M Mijares[1,2], David W Hammers[2,3], Christopher A Wolff[1,2], Neil B Wood[4], Hailey R Olafson[5], Ping Du[1], Siegfried Labeit[6], Michael J Previs[4], Eric T Wang[2,5], Karyn A Esser[1,2]\***

[1]Department of Physiology and Functional Genomics, University of Florida, Gainesville, United States; [2]Myology Institute, University of Florida, Gainesville, United States; [3]Department of Pharmacology and Therapeutics, University of Florida, Gainesville, United States; [4]Department of Molecular Physiology and Biophysics, University of Vermont, Burlington, United States; [5]Department of Molecular Genetics of Microbiology, Center for Neurogenetics, University of Florida, Gainesville, United States; [6]Medical Faculty Mannheim, University of Heidelberg, Mannheim, Germany

**\*For correspondence:**
kaesser@ufl.edu

**Present address:** [†]Foresight Diagnostics, Aurora, United States

**Competing interest:** The authors declare that no competing interests exist.

**Abstract** Circadian rhythms are maintained by a cell-autonomous, transcriptional–translational feedback loop known as the molecular clock. While previous research suggests a role of the molecular clock in regulating skeletal muscle structure and function, no mechanisms have connected the molecular clock to sarcomere filaments. Utilizing inducible, skeletal muscle specific, *Bmal1* knockout (iMS*Bmal1*[-/-]) mice, we showed that knocking out skeletal muscle clock function alters titin isoform expression using RNAseq, liquid chromatography–mass spectrometry, and sodium dodecyl sulfate-vertical agarose gel electrophoresis. This alteration in titin's spring length resulted in sarcomere length heterogeneity. We demonstrate the direct link between altered titin splicing and sarcomere length in vitro using U7 snRNPs that truncate the region of titin altered in iMS*Bmal1*[-/-] muscle. We identified a mechanism whereby the skeletal muscle clock regulates titin isoform expression through transcriptional regulation of *Rbm20*, a potent splicing regulator of titin. Lastly, we used an environmental model of circadian rhythm disruption and identified significant downregulation of *Rbm20* expression. Our findings demonstrate the importance of the skeletal muscle circadian clock in maintaining titin isoform through regulation of RBM20 expression. Because circadian rhythm disruption is a feature of many chronic diseases, our results highlight a novel pathway that could be targeted to maintain skeletal muscle structure and function in a range of pathologies.

## Editor's evaluation

Riley et al. provide a fundamental study that advances our understanding of how muscle biology is regulated by the circadian clock. The authors use compelling methodology to reveal a novel molecular mechanism for circadian regulation of the muscle giant protein titin via the splicing factor RBM20. The work will be of broad interest to muscle and circadian biologists, with implications for muscle-related disorders.

## Introduction

Circadian rhythms are intrinsically directed patterns seen in behavior and biology with an ~24 hr periodicity (*Bass and Takahashi, 2010*). Underlying circadian rhythms is a transcriptional–translational feedback loop referred to as the molecular clock. The molecular clock exists in virtually every cell in the body and functions to support homeostasis through regulating a daily pattern of gene expression (*Merrow et al., 2005*; *Yoo et al., 2004*).

BMAL1 is a PAS/basic helix–loop–helix (bHLH) transcription factor that acts as the only nonredundant component of the core clock mechanism (*Bunger et al., 2000*; *Hogenesch et al., 1998*). Beyond its role in cellular time keeping, BMAL1 and its partner, CLOCK, have been described as transcriptional regulators of both rhythmic and nonrhythmic genes important for cell-type-specific functions (*Bass and Takahashi, 2010*; *Bozek et al., 2009*; *Miller et al., 2007*). The most well-studied aspect of these functions, particularly in the context of skeletal muscle, is the circadian clock regulation of carbohydrate and lipid metabolism (*Reinke and Asher, 2019*). Our lab has provided evidence suggesting a more expansive role for the skeletal muscle clock through maintenance of skeletal muscle structure and function (*McCarthy et al., 2007*; *Andrews et al., 2010*; *Schroder et al., 2015*).

The first evidence for this role came from microarray analyses showing expression of the myogenic regulatory factor *Myod1* as well as genes encoding contractile proteins are circadian in the hindlimb muscles of C57BL/6J mice (*McCarthy et al., 2007*). The molecular clock's proposed role in maintaining muscle function was supported with whole-muscle and single-fiber mechanics data from the *Bmal1* knockout mice (*Andrews et al., 2010*). Additionally, electron microscopy studies demonstrated sarcomere irregularities consistent with the diminished force output findings. However, these experiments were performed in systemic *Bmal1* KO mice that exhibit significant behavioral abnormalities, thus, distinguishing the role of skeletal muscle molecular clock from activity patterns requires further study using more refined approaches and animal models.

Historically, sarcomeres have been studied as a two-filament system; however, a third filament comprised of the giant protein titin is now broadly recognized as having a critical role in both the structure and contractile function of sarcomeres (*Freundt and Linke, 2019*). Titin is the largest known protein and spans the length of the half-sarcomere, from Z-line to M-line, functioning as the structural backbone for the sarcomere and aiding in both active and passive force generation (*Freundt and Linke, 2019*; *Brynnel et al., 2018*; *Mijailovich et al., 2019*). The I-band region of titin connects the thick filament to the thin filament and acts as a viscoelastic spring. This region is primarily composed of immunoglobulin-like (Ig) repeats and the PEVK region. Titin undergoes a remarkable amount of splicing within this region with inclusion or exclusion of these repeats yielding longer or shorter titin proteins, respectively (*Guo et al., 2010*; *Guo et al., 2012*; *Li et al., 2013*). Maintenance of titin splicing, and thus the isoform composition, is needed to maintain muscle function.

This study utilized inducible, skeletal muscle-specific *Bmal1* (iMS*Bmal1*$^{-/-}$) knockout mice to further examine the role of the skeletal muscle molecular clock in regulating titin splicing. Importantly, these mice display muscle weakness defined by ex vivo measures of maximum isometric tetanic force (i.e., specific force) (*Schroder et al., 2015*). We show that loss of *Bmal1* specifically in adult skeletal muscle alters splicing in titin's spring region, resulting in sarcomere length disruption. Consistent with changes in splicing, we identify exon-specific protein changes in titin using a methodological approach for analyzing titin isoforms using liquid chromatography–mass spectrometry (LC-MS). Mechanistically, the skeletal muscle molecular clock regulates *Rbm20*, the gene encoding an RNA-binding protein known to regulate titin splicing. Overexpressing RBM20 in iMS*Bmal1*$^{-/-}$ skeletal muscle is sufficient to restore titin isoform expression. Use of a chronic jet-lag model with wildtype mice confirms that circadian disruption is sufficient to change Rbm20 expression. Our findings establish the role of the skeletal muscle molecular clock in regulating Rbm20 expression and titin splicing and provide a mechanism by which circadian dysregulation might alter skeletal muscle physiology.

## Results

### Titin splicing is disrupted following *Bmal1* knockout in adult skeletal muscle

Since titin has been implicated in regulating sarcomere length and plays a prominent role in striated muscle force generation, we asked whether titin protein changes following *Bmal1* knockout in skeletal muscle. Using sodium dodecyl sulfate-vertical agarose gel electrophoresis (SDS-VAGE) to analyze gross size changes in titin protein of iMS*Bmal1*$^{-/-}$ tibialis anterior (TA) muscle, we found a shift in titin isoform composition (p<0.01; *Figure 1A and B*) with no change in total titin expression. The shift in titin isoforms changes the predominant form from a short isoform to a mixture of both short and long titin proteins.

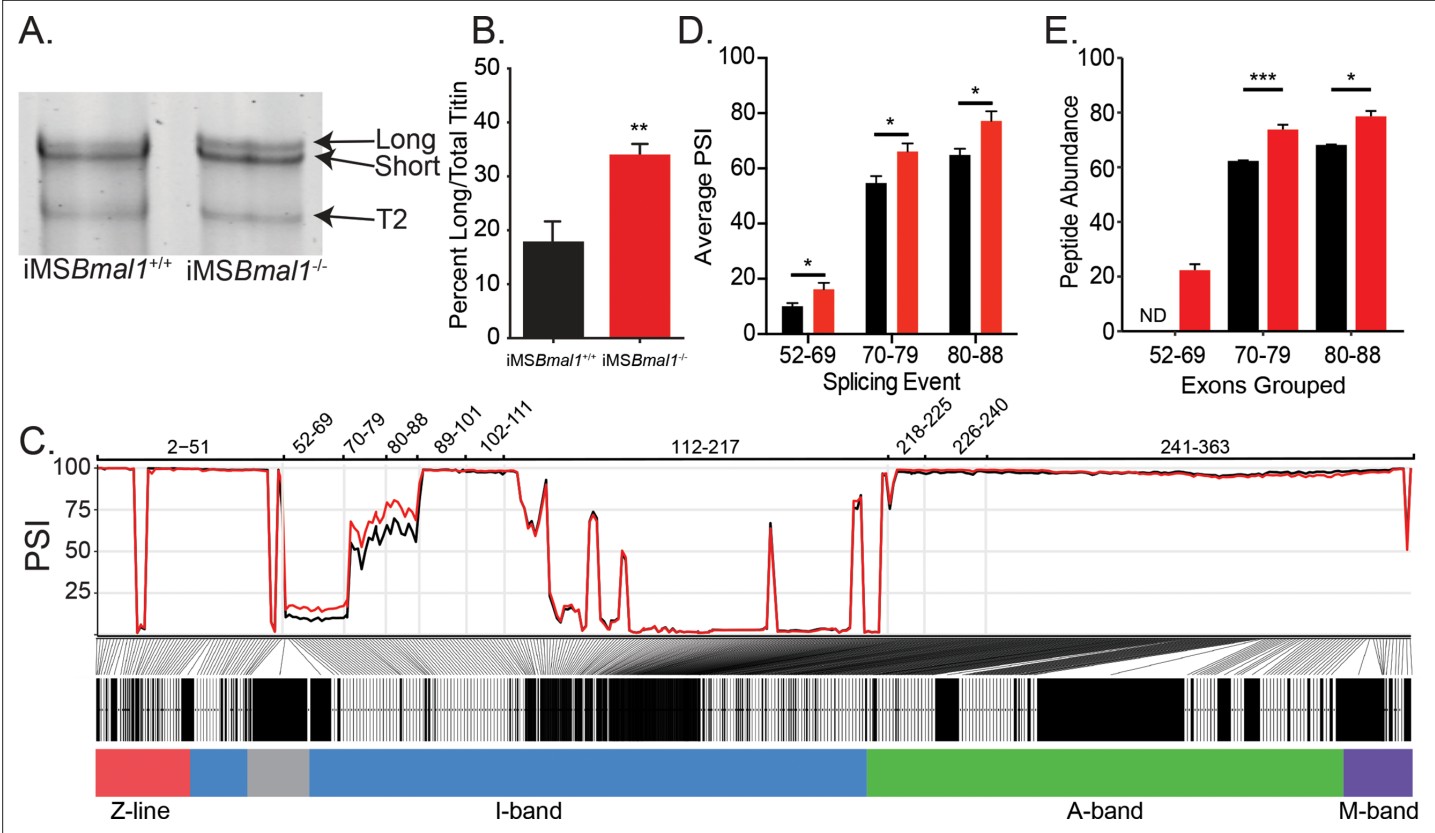

**Figure 1.** Skeletal muscle *Bmal1* knockout results in altered splicing of titin's spring region. (**A**) Sodium dodecyl sulfate-vertical agarose gel electrophoresis (SDS-VAGE) was performed to measure titin protein isoform in iMS*Bmal1*⁺/⁺ and iMS*Bmal1*⁻/⁻ tibialis anterior muscles. (**B**) Quantification of titin isoforms in these muscles shows a significant shift in titin isoform from a predominantly short isoform to a mix of long and short isoforms in iMS*Bmal1*⁻/⁻ muscles but not iMS*Bmal1*⁺/⁺ muscles (N = 8/group). (**C**) Percent spliced in (PSI) and ΔPSI of titin exons expressed in iMS*Bmal1*⁺/⁺ and iMS*Bmal1*⁻/⁻ skeletal muscle. Exons within the proximal Ig domain were included more often in iMS*Bmal1*⁻/⁻ muscle compared to iMS*Bmal1*⁺/⁺ muscle (N = 3/group). (**D**) Average PSI across three splicing events in the proximal Ig domain of tibialis anterior muscle show increased exon inclusion in iMS*Bmal1*⁻/⁻ mice compared to iMS*Bmal1*⁺/⁺ controls (N=3/group). Each event is highlighted in *Figure 2C*. (**E**) Liquid chromatography–mass spectrometry (LC-MS)-quantified peptide abundance mapping onto exons identified using RNAseq confirms changes in titin splicing are translated to titin protein (N=3/group). §No peptides were detected that mapped to exons 52–69 in iMS*Bmal1*⁺/⁺ muscle. Data plotted as mean ± SEM. Statistical significance determined by Student's t-test. *p<0.05, **p<0.01, ***p<0.0001.

The online version of this article includes the following source data for figure 1:

**Source data 1.** Sodium dodecyl sulfate-vertical agarose gel electrophoresis (SDS-VAGE) gels used for quantifying titin isoform ratios in iMS*Bmal1*⁺/⁺ and iMS*Bmal1*⁻/⁻ tibialis anterior (TA) muscle.

**Source data 2.** Titin peptide-level data used for quantifying domain-level changes to titin splicing.

Titin mRNA is well known to undergo significant number of splicing events to generate its many isoforms (*Guo et al., 2010*). To gain insight into the changes in titin isoform following loss of *Bmal1* in adult skeletal muscle, we performed RNA sequencing to identify exon-specific changes in iMS*Bmal1*⁻/⁻ muscle (*Guo et al., 2012*; *Schafer et al., 2015*; *Maatz et al., 2014*). iMS*Bmal1*⁻/⁻ TA muscle showed increased percent sliced in (PSI) of exons 52–88 of the titin transcript compared to iMS*Bmal1*⁺/⁺ control samples (*Figure 1C*). These exons code for a portion of titin's proximal Ig segment, one of titin's extensible segments within its spring region. Though the ΔPSI across this splicing event was stable, the variability in PSI across this event made statistical analysis difficult. To remedy this, we split this large splicing event into three, smaller splicing events (exons 52–69, 70–79, and 80–88) based on the similarity of PSI values within each sample (*Figure 1C*). In each of these smaller events, inclusion of exons was significantly greater in iMS*Bmal1*⁻/⁻ muscle compared to iMS*Bmal1*⁺/⁺ muscle (*Figure 1D*; p<0.05). Thus, *Bmal1* knockout in skeletal muscle altered splicing of titin's I band localized spring region.

We next asked whether this difference in *Ttn* mRNA splicing resulted in detectable differences in titin protein. To date, no paper has provided side-by-side comparisons of RNA splicing and exon inclusion of titin protein. To address this gap in the field, we homogenized skeletal muscle from iMS*Bmal1*$^{+/+}$ and iMS*Bmal1*$^{-/-}$ mice using a previously published method for analyzing MYBPC isoforms (*O'Leary et al., 2019*). Proteins were digested with trypsin, and peptide abundances were quantified by label-free LC-MS analyses. A total of 1218 peptides corresponding to the Ensembl ENSMUST00000099981.10 titin transcript were identified in the LC-MS analysis of which 1009 met our minimum threshold for quantification and were used to calculate exon-specific peptide abundance. These peptides corresponded to 225 of the 363 known exons in the *Ttn* gene (*Figure 1—source data 2*). We then analyzed differential expression of peptides and identified that those representing exons 52–69 were not detected in iMS*Bmal1*$^{+/+}$ muscle and only identified in the iMS*Bmal1*$^{-/-}$ muscle samples. The abundance of peptides from the translation of exons 70–79 and 80–88 were 18 and 16% greater in the iMS*Bmal1*$^{-/-}$ compared to the iMS*Bmal1*$^{+/+}$ muscle samples, respectively (p<0.05; *Figure 1E*). Notably, the only other region that displayed significant differences at the protein level is encoded by exons 112–217. This region is lowly expressed in both iMS*Bmal1*$^{+/+}$ and iMS*Bmal1*$^{-/-}$ muscle, thus the difference in our measurement may be a result of low read numbers across this region in the RNAseq data. These results demonstrate that the difference in *Ttn* mRNA splicing encoding exons within the proximal Ig region in the iMS*Bmal1*$^{-/-}$ muscle is translated to titin protein. Our findings from these experiments also provide a unique analytical reference for the titin research community.

## iMS*Bmal1*$^{-/-}$ muscle displays irregular sarcomere lengths

We have previously reported that the global *Bmal1* knockout mouse displays sarcomere abnormalities in skeletal muscle (*Andrews et al., 2010*); however, these sarcomeric changes could be a result of a loss of skeletal muscle *Bmal1*, a loss of behavioral rhythms, and/or other systemic interactions. Since titin acts as a sarcomeric ruler and our observed titin splicing change is present in titin protein, we measured sarcomere length in iMS*Bmal1*$^{-/-}$ skeletal muscle. To determine whether iMS*Bmal1*$^{-/-}$ skeletal muscle presents with abnormal sarcomere morphology to accompany the weakness, we longitudinally cryosectioned TA muscles and measured sarcomere lengths as the distance between peak fluorescence following α-actinin immunostaining (*Figure 2A*). Using this approach, mean sarcomere length did not significantly change following skeletal muscle-specific loss of *Bmal1* (p=0.86; *Figure 2B*). However, sarcomere length homogeneity is a feature of healthy striated muscle, and we noted that sarcomere length was significantly more variable in iMS*Bmal1*$^{-/-}$ muscle compared to iMS*Bmal1*$^{+/+}$ muscle (*F* = 22.12, p<0.05; *Figure 2B*).

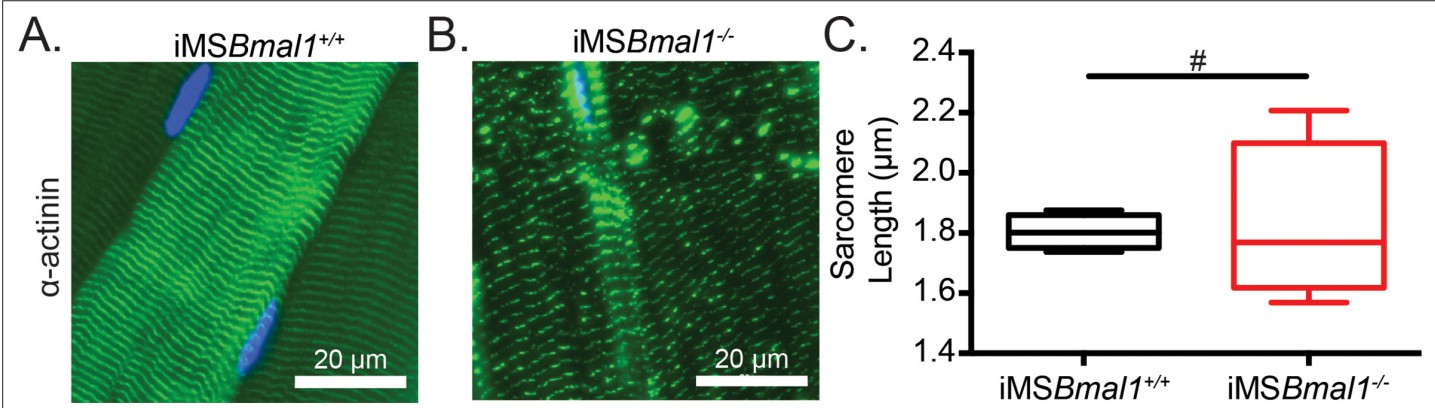

**Figure 2.** Sarcomere lengths are variable in iMS*Bmal1*$^{-/-}$ muscle. Representative images from (**A**) iMS*Bmal1*$^{+/+}$ and (**B**) iMS*Bmal1*$^{-/-}$ tibialis anterior muscles. Muscles were longitudinally cryosectioned and stained with a primary antibody against α-actinin-2. (**C**) Sarcomere lengths are significantly more variable as based on an *F*-test in iMS*Bmal1*$^{-/-}$ muscles compared to iMS*Bmal1*$^{+/+}$ controls. N = 4/group. #Significant difference in variance (p<0.05).

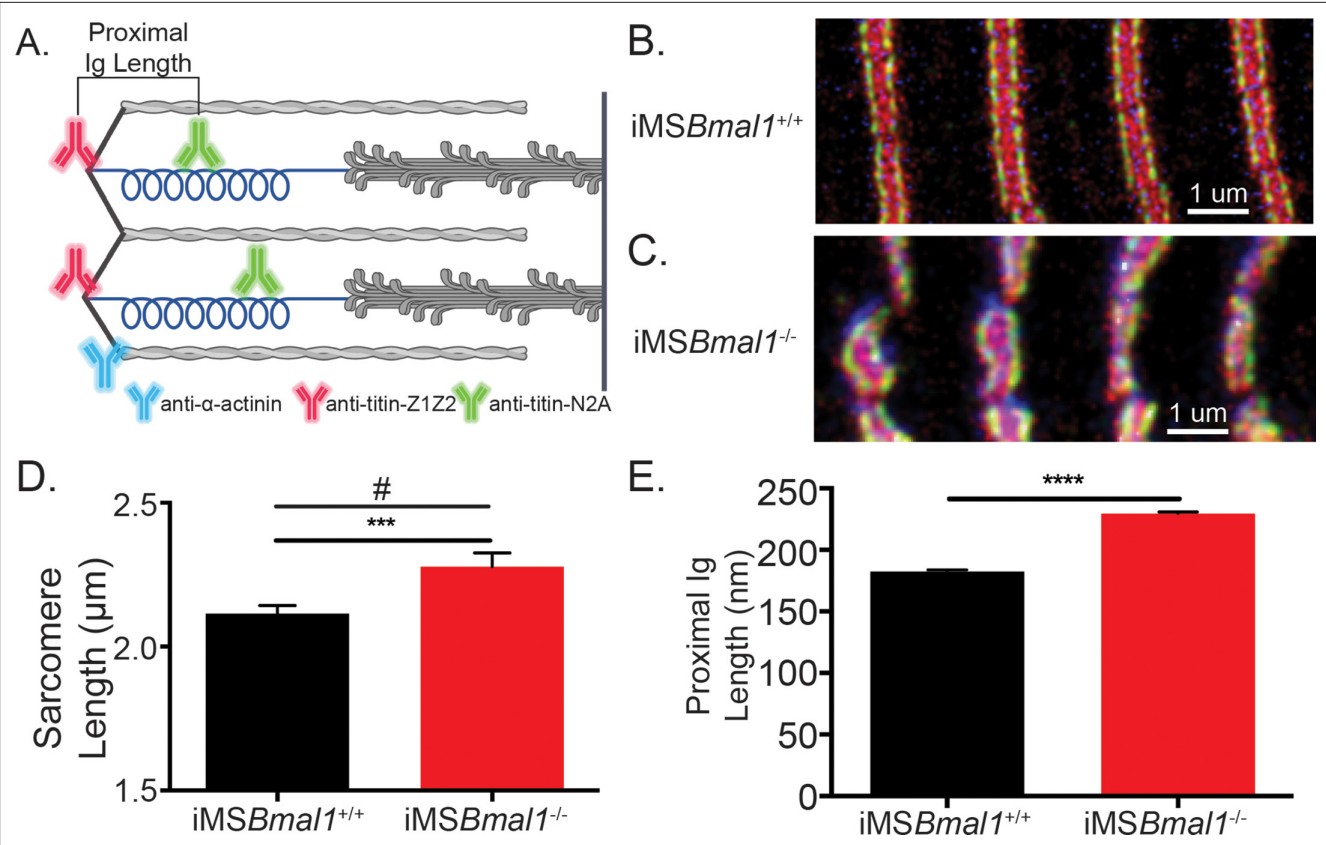

**Figure 3.** The change in titin's spring length in iMS*Bmal1*⁻/⁻ muscle accounts for sarcomere length variability. (**A**) Proximal Ig length was determined as the positional difference between the Z1Z2 domain of titin and the N2A domain of titin. These antibodies flank the proximal Ig domain. Created with BioRender.com. Representative images of immunohistochemistry using titin-epitope-specific and α-actinin-2 antibodies for (**B**) iMS*Bmal1*⁺/⁺ and (**C**) iMS*Bmal1*⁻/⁻ skeletal muscle. Green: N2A titin; red: Z1Z2 titin; blue: α-actinin-2. (**D**) Sarcomere length variability is significantly lower in iMS*Bmal1*⁺/⁺ compared to iMS*Bmal1*⁻/⁻ skeletal muscle. (**E**) A significantly longer proximal Ig length is correlated with this change in sarcomere length variability. N = 250–350 sarcomeres/three biological replicates per group. Data plotted as mean ± SEM. Statistical significance determined by Student's t-test. ***p<0.001, ****p<0.0001. # Significant difference in variance determined by F-test (p<0.05).

## Titin isoform shift accounts for sarcomere length irregularity in iMS*Bmal1*⁻/⁻ muscle

After defining the region of titin mRNA that changed in iMS*Bmal1*⁻/⁻ muscle, we asked whether splicing in this region would be sufficient to account for the increased sarcomere length variability. To test this, we performed immunohistochemistry using epitope-specific antibodies against titin's Z1Z2 and N2A domains because these epitopes flank the region with increased exon inclusion, the proximal Ig segment of titin. The positional difference between the two domains provides a proxy for proximal Ig domain length (*Figure 3A*). Representative images of iMS*Bmal1*⁺/⁺ and iMS*Bmal1*⁻/⁻ muscle can be seen in *Figure 3B and C*, respectively. The significant difference in sarcomere length variability persisted as compared through an F-test of group variance though the sarcomere lengths themselves were longer than in our previous measurements. iMS*Bmal1*⁺/⁺ muscle had sarcomere lengths of 2.12 ± 0.01 μm while iMS*Bmal1*⁻/⁻ muscle had sarcomere lengths of 2.28 ± 0.02 μm (*F* = 3.62, p<0.001; *Figure 3D*). Once we determined that sarcomere length variability was maintained, we asked whether proximal Ig domain length was significantly altered following *Bmal1* knockout in adult skeletal muscle. iMS*Bmal1*⁻/⁻ had significantly longer proximal Ig domain lengths (229.5 ± 1.357 nm) than the proximal Ig domain from iMS*Bmal1*⁺/⁺ samples (182.4 ± 1.388 nm; p<0.0001; *Figure 3E*). This difference in lengths accounts for the difference we see in sarcomere lengths in these samples. However, the variation around the mean for proximal Ig domain length was not significantly different (*F* = 1.338, p=0.12). These changes confirm that the proximal Ig domain length in adult skeletal muscle is correlated with a corresponding change in sarcomere length.

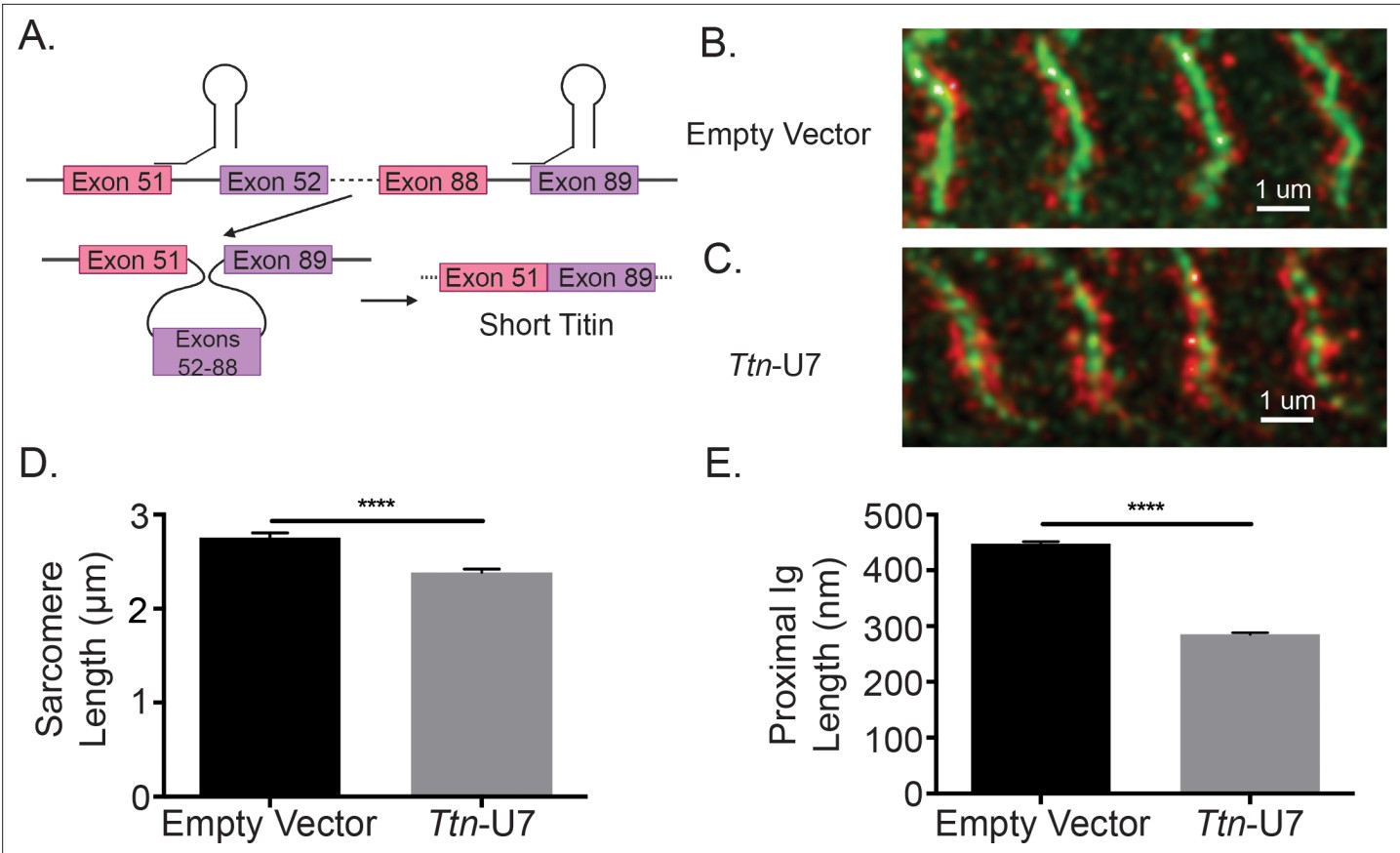

**Figure 4.** Directly shortening titin's proximal Ig domain length results in shorter sarcomeres in vitro. (**A**) U7 snRNPs were designed to induce splicing or the proximal Ig domain with dysregulated splicing in iMS*Bmal1*[-/-] muscle. Created with BioRender.com. Representative images of immunocytochemistry using a titin-epitope-specific antibody and stably expressing eGFP-ACTN2 C2C12 myotubes that were transfected with an (**B**) empty vector control or (**C**) *Ttn*-U7 splicing factors. Green: α-actinin-2; red: N2A titin. (**D**) Sarcomere lengths were significantly shorter in myotubes transfected *Ttn*-U7 vectors compared to empty vector controls. (**E**) *Ttn*-U7 vector transfection resulted in significantly shorter proximal Ig domain lengths in eGFP-ACTN2-C2C12 myotubes. N = 100-200 sarcomeres/two biological replicates per group. Data plotted as mean ± SEM. Statistical significance determined by Student's t-test. ****p<0.0001.

## Changes in titin I-band splicing are directly linked to changes in sarcomere length

While the changes to sarcomere length of iMS*Bmal1*[-/-] TA muscle are associated with titin splicing and length, this model does not causally link titin splicing within the proximal Ig region to sarcomeric structure. Thus, we thought it necessary to manipulate titin splicing without the influence of the clock. To directly test whether the region of titin that is altered in iMS*Bmal1*[-/-] muscle could contribute to changes in sarcomere length independent of clock changes, we used U7 snRNPs to direct titin splicing from the 5'splice site of titin exon 51 to the 3'splice site of titin exon 89, resulting in shorter titin protein using an in vitro C2C12 myotube model. To visualize sarcomere length, we used a stable cell line in which α-actinin is labeled with GFP (eGFP-ACTN2-C2C12). These labeled myocytes were transfected with the titin-specific U7 snRNPs and plated on micropatterned gelatin hydrogels to support maturation of the myotube culture and sarcomere formation and alignment (*Denes et al., 2019*; *Bettadapur et al., 2016*). At day 10 of differentiation, eGFP-ACTN2-C2C12 myotubes were fixed then stained with a titin N2A epitope-specific antibody similarly to the tissue samples from iMS*Bmal1*[-/-] muscle (*Figure 4A*). Representative images of control eGFP-ACTN2-C2C12 myotubes and *Ttn*-U7 transfected eGFP-ACTN2-C2C12 myotubes can be seen in *Figure 4B and C*, respectively. While control myotubes had an average sarcomere length of 2.756 ± 0.0495 µm, *Ttn*-U7 transfected myotubes had significantly shorter sarcomeres with an average sarcomere length of 2.387 ± 0.0347 µm (p<0.0001, n = 100–200 sarcomeres/two biological replicates; *Figure 4D*). Because

ACTN2 is a key component of the Z-line, the distance between the N2A-epitope and eGFP-ACTN2 was used to calculate proximal Ig length in this experiment. Proximal Ig domain lengths were also significantly different. *Ttn*-U7 transfected myotubes had a proximal Ig length of 285.1 ± 3.131 nm compared to 448.0 ± 3.368 nm of control myotubes (p<0.0001; *Figure 4E*). These results confirm that altering the length of titin's proximal Ig domain causes a change in sarcomere length in skeletal muscle myotubes.

## Rbm20 is targeted by the skeletal muscle molecular clock

Having established that iMS*Bmal1*[-/-] skeletal muscle has an altered titin isoform composition resulting in sarcomere length heterogeneity, we looked to define the mechanism linking the skeletal muscle molecular clock to titin splicing. RBM20 is a well-characterized splicing factor known to target titin pre-mRNA (*Guo et al., 2012*; *Li et al., 2013*; *Maatz et al., 2014*). RBM20 is an RNA-binding protein that acts as a splicing repressor such that when present at high levels it binds along titin pre-mRNA preventing normal intronic splicing. When bound by many RBM20 proteins, titin pre-mRNA undergoes large exon-skipping events whereby the resultant, processed mRNA is shorter than if RBM20 was not present. We tested whether RBM20 levels change following loss of skeletal muscle *Bmal1*. *Rbm20* mRNA expression was reduced by 34% in iMS*Bmal1*[-/-] muscle compared to iMS*Bmal1*[+/+] muscle (p<0.05; *Figure 5A*). This reduced expression was also present at the protein level where RBM20 protein content decreased by 46% compared to iMS*Bmal1*[+/+] muscle (p<0.05; *Figure 5B*).

We next used an environmental model to chronically disrupt the expression of the muscle clock to ask whether this was sufficient to change *Rbm20* expression in muscle. C57BL/6J mice were subjected to 8 weeks of repeated phase advances, a well-established model of circadian disruption most commonly referred to as chronic jet lag. Prior studies using this model demonstrated significant disruption of circadian clock gene expression in skeletal muscle (*Wolff et al., 2013*). For this analysis, we included collections at two time points so that we could capture potential changes during either the rest (CT30) or active (CT42) parts of the day. In *Figure 5D*, we found that Rbm20 mRNA did exhibit a significant difference in expression in the quadriceps muscles under normal light:dark conditions. In contrast, muscles from the CPA mice exhibit depressed expression of Rbm20, mostly notably at CT30, during the subjective rest phase for the mice. The results for the TA muscle from the same mice follow a similar trend with reduced Rbm20 expression at both time points in the CPA group, but this change was not significant (p=0.09). The observation that Rbm20 mRNA expression was differentially expressed in the rest vs. active period of the quadriceps muscles was not expected but is consistent with the muscle clock regulating Rbm20 expression. In summary, these data indicate that *Rbm20* expression is regulated downstream of the skeletal muscle circadian clock; this regulation is evident in two different muscles, and it provides further evidence that altering the muscle clock, either genetically or physiologically, leads to altered *Rbm20* expression.

Since skeletal muscle *Bmal1* knockout and physiological disruption of circadian rhythms reduced *Rbm20* expression, we asked whether *Rbm20* is a direct transcriptional target of the core clock factors in skeletal muscle. Our first approach was to interrogate our skeletal muscle ChIP-seq data sets for BMAL1 and CLOCK as these proteins comprise the positive limb of the skeletal muscle circadian clock (*Gabriel et al., 2021*). We also looked into our skeletal muscle ChIP-seq data set for MYOD1 as we have previously shown with *Tcap* that the clock factors can function synergistically with MYOD1 to transcriptionally regulate gene expression (*Hodge et al., 2019*; *Cao et al., 2010*). As seen in *Figure 5C*, we detected significant binding of BMAL1, CLOCK, and MYOD1 within a common region of intron 1 of *Rbm20*. Analysis of this region identified a putative regulatory element as defined by the third phase of the ENCODE project (*Moore et al., 2020*). To confirm binding of these proteins, we performed targeted ChIP-PCR using primers flanking this 88 bp site. We identified significant binding of BMAL1 (p<0.01), CLOCK (p<0.001), and MYOD1 (p<0.001; *Figure 5F*). To test whether this site served as an enhancer for *Rbm20* transcription in muscle cells, we generated luciferase reporter constructs with either the *Rbm20* proximal promoter alone (*Rbm20*-Luc) or the ~400 bp enhancer region from intron 1 cloned upstream of the *Rbm20* promoter (E-*Rbm20*-Luc). We found that over-expression of the clock factors in E-*Rbm20*-Luc transfected myotubes resulted in a 255-fold increase in luciferase activity compared to empty vector control transfected myotubes (n = 3, p<0.0001, *Figure 5G*). Importantly, this increase is 9.8-fold higher than the increase seen in myotubes transfected with *Rbm20*-P-Luc, which lacks the enhancer element (p<0.0001). These data have identified a

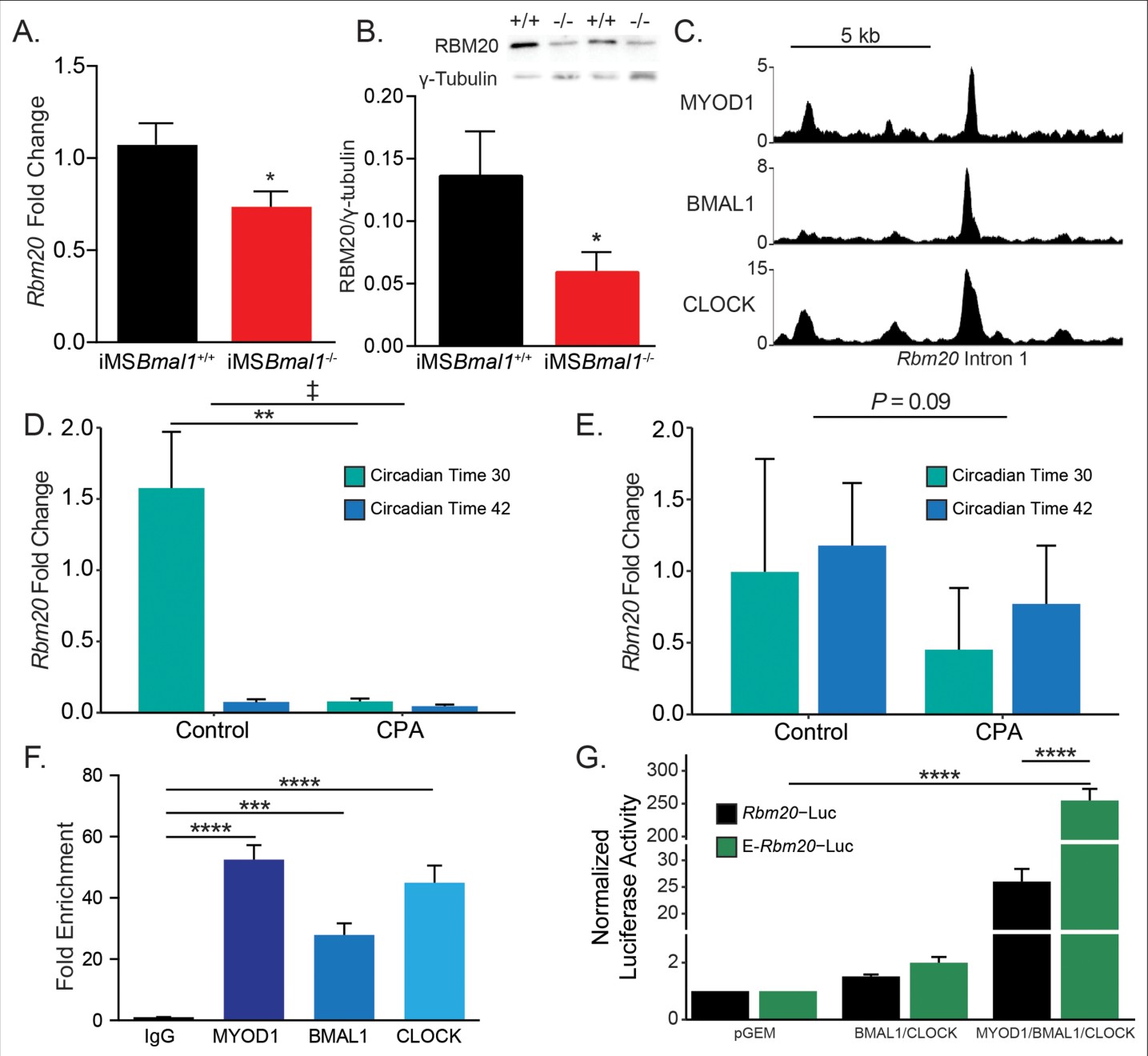

**Figure 5.** *Rbm20* is targeted by the positive limb of the skeletal muscle molecular clock.

(**A**) *Rbm20* mRNA is decreased by 34% in iMS*Bmal1*^-/- muscle compared to iMS*Bmal1*^+/+ muscle (N = 6–8/group). (**B**) iMS*Bmal1*^-/- muscle shows a 57% reduction in RBM20 protein levels compared to iMS*Bmal1*^+/+ control muscle (N = 4/group). (**C**) BMAL1, CLOCK, and MYOD1 ChIP-seq data were used to identify potential regulatory regions for *Rbm20*. A significant binding peak was found within intron 1 of this gene. (**D**) C57BL/6J mice subjected to repeated phase advances (CPA) show a significant reduction in *Rbm20* mRNA expression in the quadriceps muscle (N = 4/group). (**E**) C57BL/6J mice subjected to CPA show a nonsignificant reduction in *Rbm20* expression in the tibialis anterior muscle (N = 3/group). (**F**) ChIP-PCR confirms binding of each protein of the positive limb of the molecular clock at this site (N = 3/group). (**G**) Dual-luciferase activity shows significant activation of E-*Rbm20*-Luc with overexpression of the positive limb of the molecular clock (N = 3/group). Data plotted as mean ± SEM. Statistics performed using Student's t-test or one-way ANOVA with Tukey's post-hoc test. *p<0.05, **p<0.01, ****p<0.0001, ‡ Group effect p<0.05.

The online version of this article includes the following source data for figure 5:

**Source data 1.** RBM20 expression is decreased in iMS*Bmal1*^-/- muscle compared to iMS*Bmal1*^+/+ muscle.

novel circadian clock-sensitive enhancer in the *Rbm20* gene and support a model in which the skeletal muscle molecular clock modulates *Rbm20* transcription through this intronic enhancer.

## RBM20 overexpression in iMS*Bmal1*$^{-/-}$ muscle restores titin isoform expression

Since *Rbm20* expression is regulated by the skeletal muscle molecular clock, we tested whether over-expression of RBM20 expression, using direct muscle injection of *Rbm20*-AAV, in the iMS*Bmal1*$^{-/-}$ muscle would be sufficient to restore titin splicing and protein uniformity (**Figure 6A**). Consistent with our prior measures, RBM20 protein levels decreased by 38% in iMS*Bmal1*$^{-/-}$ muscle compared to iMS*Bmal1*$^{+/+}$ muscle injected with GFP-AAV ($p<0.05$). Injection of RBM20-AAV into iMS*Bmal1*$^{-/-}$ muscle increased *Rbm20* gene expression by 2.8-fold ($p_{adj}<0.05$) and RBM20 protein expression almost 2-fold above iMS*Bmal1*$^{+/+}$-GFP-AAV levels ($p<0.05$; **Figure 6B**). We performed SDS-VAGE with these muscles to determine titin isoform distribution. Consistent with earlier work, loss of muscle *Bmal1* resulted in a significant increase in the long form of titin ($p<0.01$; **Figure 6C**). AAV expression of RBM20 in the iMS*Bmal1*$^{-/-}$ muscle was sufficient to significantly reduce the amount of the long titin isoform to levels closer to wildtype ($p<0.05$; **Figure 6C**). These findings demonstrate that the decrease in RBM20 expression downstream of loss of muscle *Bmal1* contributes significantly to titin isoform heterogeneity in skeletal muscle.

While titin protein size is partially rescued in RBM20-overexpressing iMS*Bmal1*$^{-/-}$ muscle, whether or not the proximal Ig domain length of titin is restored remained undetermined. We performed RNAseq on iMS*Bmal1*$^{-/-}$-GFP and iMS*Bmal1*$^{-/-}$-RBM20 TA muscle to analyze the splicing of the titin transcript. iMS*Bmal1*$^{-/-}$-RBM20 muscle showed decreased expression of exons 70–88 of titin compared to iMS*Bmal1*$^{-/-}$-GFP muscle (**Figure 6D**). Importantly, these exons are increased in iMS*Bmal1*$^{-/-}$ muscle compared to iMS*Bmal1*$^{+/+}$ muscle, suggesting that these exons are regulated downstream of the molecular clock through *Rbm20* expression. We did note, however, that exons 52–69 showed only a modest increase in expression in muscle-overexpressing RBM20. This change was unexpected and is likely a result of other splice factors or mechanisms regulating inclusion/exclusion of those exons. These changes were confirmed using LC-MS with a significant decrease in exons 70–79 ($p<0.001$) and 80–88 ($p<0.01$) inclusion in iMS*Bmal1*$^{-/-}$-RBM20-AAV TA muscle with no significant different in inclusion of exons 52–69 in titin protein (**Figure 6E**). We used IHC to probe whether rescue of Rbm20 expression in the iMS*Bmal1*$^{-/-}$ muscle results in a change in the distance from the Z-disc to the PEVK region of titin. Longitudinal sections were labeled for these two epitopes (α-actinin/TTN-PEVK) demonstrated that RBM20 rescue with rescue of TTN-splicing resulted in a reduced distance of the PEVK domain from the Z-disc, consistent with a smaller titin isoform within the sarcomere (**Figure 6—figure supplement 1**). Unfortunately, we only had a small amount of tissue remaining, so our sample size is small, but these results are consistent with the levels of the level of exon inclusion in both the RNAseq and LC-MS data,.thus, demonstrating that splicing of titin is modulated downstream of the circadian clock via regulation of Rbm20 expression.

## Discussion

In this study, we examined the contribution of the core clock factor *Bmal1* on regulation of titin isoform expression in skeletal muscle. We defined a novel mechanism whereby the skeletal muscle clock regulates titin splicing through modulation of *Rbm20* expression. These findings define that skeletal muscle structure, with implication for function, is regulated in part through a circadian clock-dependent transcriptional mechanism (**Miller et al., 2007**; **McCarthy et al., 2007**; **Andrews et al., 2010**; **Schroder et al., 2015**; **Hodge et al., 2019**). The novel link between the muscle circadian clock and RBM20 is of particular importance as our lab recently proposed a mechanism whereby MYOD1 works with BMAL1::CLOCK to cooperatively regulate skeletal muscle-specific, clock-controlled genes (**Hodge et al., 2019**). Here, we identify a novel enhancer element within intron 1 of *Rbm20* and provide evidence that MYOD1 with the core clock factors are significant contributors to *Rbm20* expression. These data confirm that *Rbm20* is a bona fide clock-controlled gene in skeletal muscle. We provide direct evidence that changes to the proximal Ig segment of titin's spring region can alter sarcomere length and further support titin's role as a molecular ruler. Further, we provide gain of function transcriptome and proteome data, which confirm that RBM20

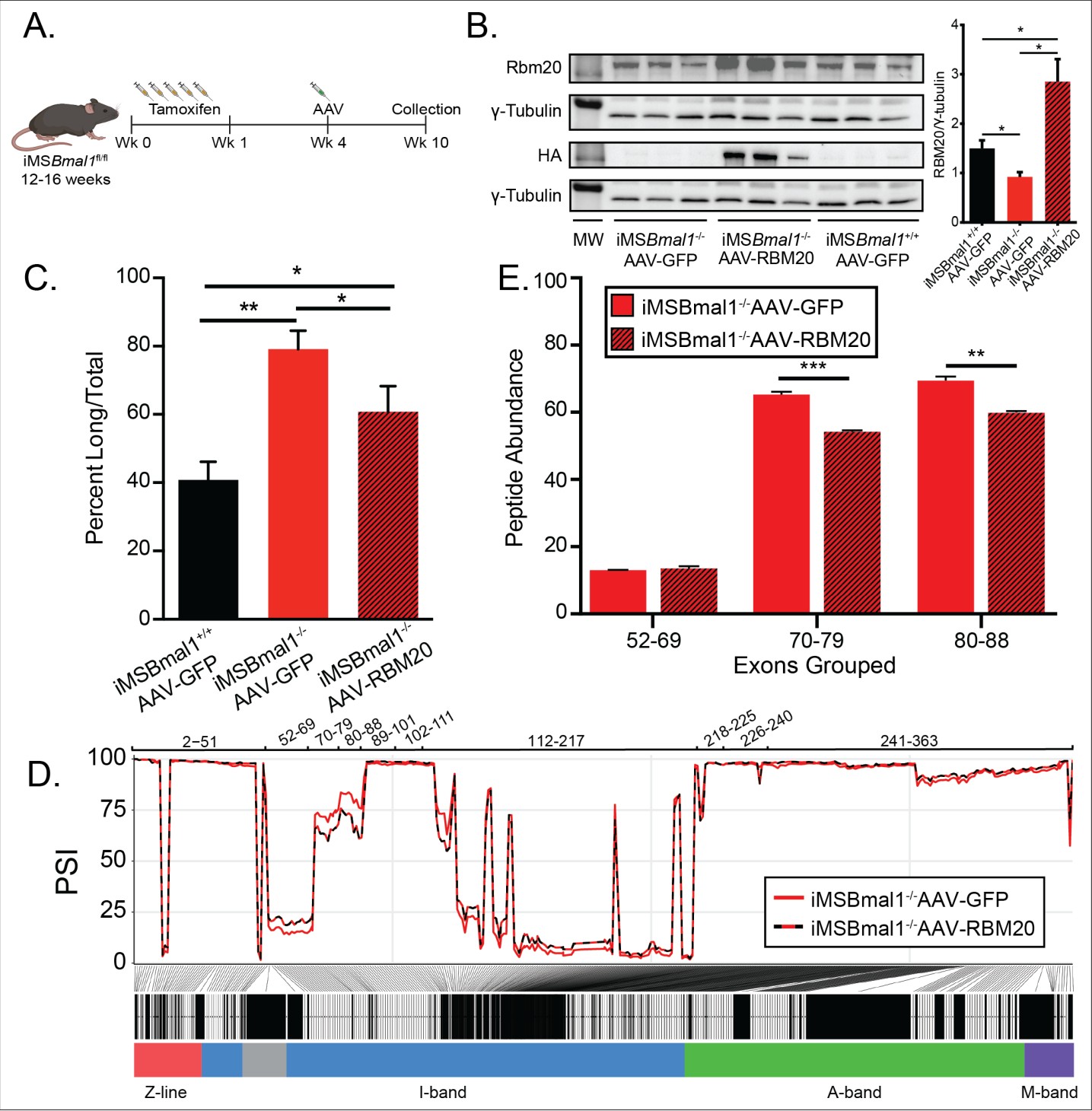

**Figure 6.** Rescuing RBM20 expression in iMS*Bmal1*-/- muscle restores titin spring length and sarcomere length variability. (**A**) Protocol for conditionally knocking out skeletal muscle Bmal1 and overexpressing RBM20. (**B**) Transduction of AAV-RBM20 in iMS*Bmal1*-/- muscle increases RBM20 91% over iMS*Bmal1*+/+ AAV-GFP muscle and threefold over iMS*Bmal1*-/- AAV-GFP muscle (N = 4/group). Molecular weight markers (MW) correspond to 150 kDa in RBM20 and HA blots and 50 kDa in tubulin blots. (**C**) Titin isoform expression is partially restored in iMS*Bmal1*-/- AAV-RBM20 muscle (N = 4/group). (**D**) Percent spliced in (PSI) of every exon of *Ttn* between iMS*Bmal1*-/--GFP and iMS*Bmal1*-/--RBM20 muscle (N = 3/group). (E) Liquid chromatography–mass spectrometry (LC-MS)-quantified peptide abundance mapping onto exons identified using RNAseq confirms changes in titin splicing are translated to titin protein (N=3/group). Data plotted as mean ± SEM. Statistics performed using Student's t-test or one-way ANOVA with Tukey's post-hoc test. *p<0.05; **p<0.01.

The online version of this article includes the following source data and figure supplement(s) for figure 6:

*Figure 6 continued on next page*

*Figure 6 continued*

**Source data 1.** RBM20 and HA protein expression are increased in iMS*Bmal1*[-/-] AAV-RBM20 muscle lysates compared to iMSBmal1[-/-] AAV-GFP and iMSBmal1[+/+] AAV-GFP.

**Figure supplement 1.** Rescue of RBM20 protein expression in iMS*Bmal1*[-/-] muscle results in a reduction in the distance of the titin-PEVK domain from the sarcomere Z-disc.

is a key mediator of titin splicing downstream of the circadian clock. Our findings expand upon the importance of the molecular clock in maintaining skeletal muscle health by showing that circadian rhythms are not only important for the metabolic function but are also important for maintaining muscle structural integrity.

Our results identified a new and direct link between the core circadian clock in skeletal muscle with the highly conserved splicing repressor, RBM20. RBM20 is predominantly expressed in cardiac and skeletal muscles. Because most RBM20 research has been in cardiac muscle as RBM20 mutations are linked to cardiomyopathies (*Guo et al., 2010*), understanding of its role in skeletal muscle remains limited (*Guo et al., 2010*). Our work has identified a novel enhancer element within intron 1 of *Rbm20* that confers sensitivity to MYOD1 and the core clock factors BMAL1, CLOCK. This region of the *Rbm20* gene is highly conserved and is maintained in the human ENCODE database of putative enhancers (E1499168). It is also important to note that RBM20 is currently linked to splicing of up to 18 mRNAs in the heart (*Maatz et al., 2014*), 5 of which have significant changes in exon inclusion in our model. Thus, regulation of RBM20 downstream of the circadian clocks could contribute to other aspects of muscle structure and function independent of titin splicing. It is important to note that there is a lack of overlap between the RBM20-dependent splicing changes in the heart and our model in skeletal muscle, suggesting that RBM20 may have tissue-specific targets. This tissue specificity warrants further study. Since circadian rhythm disruption occurs in aging and many chronic diseases, our findings suggest RBM20 and titin splicing could play a broader role in conditions with known muscle weakness.

Defining mechanisms to preserve muscle function is important due to the emerging recognition that maintaining muscle strength in chronic disease states and aging is a strong predictor of mortality (*Newman et al., 2006*; *García-Hermoso et al., 2018*). Since skeletal muscle function is tightly linked to its structure, we investigated a mechanism that links the molecular clock to sarcomere organization in skeletal muscle. One of the findings of the this study is an increase in sarcomere length heterogeneity following skeletal muscle *Bmal1* knockout. Heterogeneous sarcomere lengths would be disadvantageous at the muscle tissue level as the likelihood that an individual sarcomere would be operating at its optimal length is reduced (*Hou, 2018*). While a sarcomere can generate force outside of optimal length, it is well established that when the number of crossbridges between the thin and thick filaments is reduced, the maximum force produced by the muscle is diminished (*Patel et al., 2004*). Additionally, emerging data suggests that changes to overall titin isoform in skeletal muscle result in changes to the number of serial sarcomeres in an apparent effort to reduce the working range of the sarcomere, with functional consequences. However, these data come from models in which the change in titin isoform is a result of large-scale exon deletions, resulting in a homogenous expression of mutant titin. The effects of heterogenous changes in titin isoform expression, as seen in iMS*Bmal1*[-/-], with regard to sarcomere length homogeneity and working range, need further study to understand its functional consequence.

Lastly, our study used RNAseq analysis to define the titin splicing changes downstream of the muscle clock. We then determined whether the splicing changes were directly linked to changes in the titin protein using LC-MS. To our knowledge, we are the first group to provide direct correlations between titin exon splicing at the RNA level with peptide expression at the protein level for the giant protein. Notably, analysis of titin protein isoforms has historically required the use of vertical agarose gels and antibodies that detect repetitive elements to semi-quantitatively measure changes in alternative splicing. While these methods are useful, they cannot provide a direct, quantitative measurement of exon-level translation. We successfully used LC-MS to confirm that peptides encoded titin exons that undergo differential splicing in iMS*Bmal1*[-/-] mice as measured by RNAseq were similarly changed in titin protein. Our methodology could be used to confirm changes in other conditions where titin splicing is disrupted.

## Ideas and speculation

Why the molecular clock would regulate splicing of a protein as large as titin is puzzling. Estimations of titin turnover have been measured on the order of days, much longer than processes that would predictably be regulated daily by the circadian clock (*Isaacs et al., 1989*). However, studies in cardiomyocytes suggest that sarcomeric titin exhibits dynamic exchange within ~14 hr independent of protein turnover (*da Silva Lopes et al., 2011*). If such a mechanism exists in skeletal muscle, this would allow the fiber to modulate titin length on a more frequent, potentially daily basis. Previous studies have identified titin-cap (*Tcap*) as a direct clock-controlled gene that exhibits time-of-day variations in protein amount with highest levels during the active phase (*Gregorio et al., 1998*; *Hammouda et al., 2012*). TCAP functions to anchor the titin filament at the Z-line by binding titin's Z1Z2 domain (*Lee et al., 2006*; *Gregorio et al., 1998*). These observations suggest a model in which titin protein could be exchanged daily, likely in sync with times at which TCAP protein is lowest. This daily replacement would serve to provide flexibility in the muscle fibers to adjust to patterns of use and disuse. Further, this daily replacement could lead to a vulnerable period for skeletal muscle damage (i.e., when TCAP is low and titin is being replaced). The potential for this period of vulnerability is somewhat supported by studies that measured diurnal variation in muscle damage following maximum performance tests in athletes. In these studies, muscle creatine kinase (*Hammouda et al., 2012*) and leukocyte infiltration (*Hammouda et al., 2011*) increased more substantially following evening exertion compared to morning exertion. In our proposed scenario, sarcomeric replacement during the inactive period would leave skeletal muscle susceptible to damage in the evening (i.e., leading into the inactive period), thus damage markers should increase at this stage. Studies that measure sarcomeric replacement over time of day with differing time of exercise bouts should be performed to directly test this scenario as they could prove useful for prescribing exercise as a therapeutic in cases of circadian rhythm disruption.

## Conclusion

This study uniquely links the field of circadian rhythms with well-established properties of skeletal muscle structure and function through the RNA-binding protein RBM20. Since maintenance of circadian rhythms and skeletal muscle strength is gaining recognition for their roles in human health, understanding these processes is needed to promote health. Our work provides important new insights into understanding fundamental mechanisms regulating muscle homeostasis and potential new targets through which diseases that modulate the circadian clock could lead to skeletal muscle weakness.

# Materials and methods

**Key resources table**

| Reagent type (species) or resource | Designation | Source or reference | Identifiers | Additional information |
|---|---|---|---|---|
| Genetic reagent (*Mus musculus*) | iMS*Bmal1*fl/fl | https://doi.org/10.1186/s13395-015-0039-5 | | Tamoxifen-inducible, skeletal muscle-specific deletion *Bmal1* resulting in lack of BMAL1 in this tissue |
| Antibody | Anti-sarcomeric α-actinin (rabbit monoclonal) | Abcam | EP2529Y | (1:1000) |
| Antibody | Anti-titin N2A (rabbit polyclonal) | Myomedix | TTN-4 | (1:250) |
| Antibody | Anti-titin Z1Z2 (rabbit polyclonal) | Myomedix | TTN-1 | (1:100) |
| Antibody | Anti-RBM20 (rabbit polyclonal) | Myomedix | RBM20-1 | (1:500) |
| Antibody | Anti-PEVK (rabbit polyclonal) | Myomedix | PEVK-1 | (1:500) |
| Antibody | Anti-γ-tubulin (mouse monoclonal) | Sigma-Aldrich | T6557 | (1:1000) |

*Continued on next page*

*Continued*

| Reagent type (species) or resource | Designation | Source or reference | Identifiers | Additional information |
|---|---|---|---|---|
| Antibody | Anti-HA High Affinity (rat monoclonal IgG1) | Roche | 11867423001 | (1:1000) |
| Antibody | Alexa Fluor-488 conjugated goat anti-rabbit IgG (goat polyclonal) | Thermo Scientific | A11034 | (1:500) |
| Antibody | Alexa Fluor-Plus 405 conjugated goat anti-rabbit IgG (goat polyclonal) | Thermo Scientific | A48254 | (1:500) |
| Antibody | Alexa Fluor-647 conjugated goat anti-rabbit IgG (goat polyclonal) | Thermo Scientific | A21244 | (1:500) |
| Antibody | HRP conjugated goat anti-rabbit IgG (H+L) (goat polyclonal) | Sigma-Aldrich | AO307P | (1:10,000) |
| Antibody | HRP conjugated goat anti-mouse IgG (H+L) (goat polyclonal) | Sigma-Aldrich | 401215 | (1:10,000) |
| Sequence-based reagent | U7-BglII F | https://doi.org/10.1007/978-1-61737-982-6_11 | PCR primers | GGGAGATCTTTAACAACATAGGAGCTGTGATTGGCTGT |
| Sequence-based reagent | U7-PstI R | https://doi.org/10.1007/978-1-61737-982-6_11 | PCR primers | AAACTGCAGCACAACGCGTTTCCTAGGAAACCA |
| Sequence-based reagent | SDM_U7 smOpt F | http://doi.org/10.1007/978-1-61737-982-6_11 | PCR primers | GCTCTTTTAGAATTTTTGGAGCAGGTTTTCTGAC |
| Sequence-based reagent | SDM_U7 smOpt R | https://doi.org/10.1007/978-1-61737-982-6_11 | PCR primers | GTCAGAAAACCTGCTGGTTAAATTCTAAAAGAGC |
| Sequence-based reagent | *Ttn*-51-AS F | This paper | PCR primers | TTAGGGTGGGTGGATACGCCTCTGC AAAAGAATTTTTGGAGCAGGTTTTCTG |
| Sequence-based reagent | *Ttn*-51-AS R | This paper | PCR primers | TATCCACCCACCCTAAGTCCCTATCATAGCGGAAGTGCGTCTGTAG |
| Sequence-based reagent | *Ttn*-89-AS F | This paper | PCR primers | TAGGGTGCAAGGTACTCCTTAGAGTGAAAGAATTTTTGGA GCAGGTTT |
| Sequence-based reagent | *Ttn*-89-AS R | This paper | PCR primers | GTACCTTGCACCCTAAGTCCCTATCATAGCGGAAGTGCGTCTGTAG |
| Sequence-based reagent | *Rbm20* qPCR F | https://doi.org/10.1161/CIRCULATIONAHA.117.031947 | PCR primers | TGCATGCCCAGAAATGCCTGCT |
| Sequence-based reagent | *Rbm20* qPCR R | https://doi.org/10.1161/CIRCULATIONAHA.117.031947 | PCR primers | AAAGGCCCTCGTTGGAATGGCT |
| Sequence-based reagent | *Rpl26* qPCR F | https://doi.org/10.7554/eLife.43017 | PCR primers | CGAGTCCAGCGAGAGAAGG |
| Sequence-based reagent | *Rpl26* qPCR R | https://doi.org/10.7554/eLife.43017 | PCR primers | GCAGTCTTTAATGAAAGCCGTG |

*Continued on next page*

*Continued*

| Reagent type (species) or resource | Designation | Source or reference | Identifiers | Additional information |
|---|---|---|---|---|
| Sequence-based reagent | *Rbm20* Intron 1 F | This paper | PCR primers | CTAGGACGGAATCTGCTGTG |
| Sequence-based reagent | *Rbm20* Intron 1 R | This paper | PCR primers | AACAGGGTGTCTGTCTGTCT |
| Cell line (*M. musculus*) | eGFP-ACTN2-C2C12 | https://doi.org/10.1186/s13395-019-0203-4 | | Allows for visualization of sarcomeres in live C2C12 myotubes |
| Commercial assay or kit | Dual-luciferase reporter assay system | Promega | E1960 | |
| Commercial assay or kit | X-tremeGENE 9 DNA transfection reagent | Roche | 6365787001 | |
| Commercial assay or kit | QuikChange II Site-Directed Mutagenesis kit | Agilent | 200523 | |
| Software, algorithm | Prism 7 | GraphPad | | www.graphpad.com; |
| Software, algorithm | HISAT2 | https://doi.org/10.1038/nprot.2016.095 | | http://daehwankimlab.github.io/hisat2/ |
| Software, algorithm | R; RStudio | R Project for Statistical Computing; RStudio | | www.r-project.org; www.rstudio.com |

## Animal studies

The iMS*Bmal1* mouse model has been described (*Schroder et al., 2015*; *Hodge et al., 2015*). The *Bmal1*lox mouse was purchased from Jackson Laboratories (stock number 7668) and crossed with the skeletal muscle-specific Cre-recombinase mouse (HSA-Cre) generated in-house (*McCarthy et al., 2012*). All experiments were performed in male mice housed in an AAALAC-regulated facility under 12 hr light–12 hr dark cycle with ad libitum food and water availability. Breeding for iMS*Bmal1*fl/fl mice was performed as described in *Hodge et al., 2015*, and recombination was induced in 12–16-week-old male mice through intraperitoneal vehicle (15% ethanol in sunflower seed oil) or tamoxifen injections (2 mg day⁻¹) for 5 days (*Hodge et al., 2015*). Five weeks after injections were completed, mice were sacrificed through cervical dislocation and tissues were collected. Tissues were flash-frozen in liquid nitrogen and stored at –80°C for RNA and protein analysis. All animal procedures were approved by the University of Kentucky's and University of Florida's Institutional Animal Care and Use Committees.

For the chronic phase advance (CPA) model, 12–16-week-old C57/Bl6J mice were individually housed in 12L:12D conditions in temperature-, humidity-, and light-controlled circadian cabinets (Actimetrics, Wilmette, IL). After 2 weeks of acclimation to individual housing, mice were randomly assigned to either control (12L:12D) conditions or a CPA regimen. The CPA paradigm consisted of a 6 hr advance in lights-on every 4 days for a total of 8 weeks. This protocol was chosen as our lab has previously shown that this protocol disrupts the core molecular clock rhythm in skeletal muscle (*Wolff et al., 2013*). Disruptions in activity rhythms were confirmed using wheel-running activity. At the end of the 8-week experimental protocol, mice were put into constant darkness then sacrificed under red light at either circadian time (CT) 30 or CT42. For reference, CT30 reflects a sample collected in the middle of the rest phase for the mice and CT42 is the middle of the active phase. Muscles were collected and snap-frozen in liquid nitrogen prior to processing.

## Immunohistochemistry

TA muscles were pinned at slack length on a cork block, flash-frozen in OCT compound (Tissue-Tek) in liquid nitrogen-cooled isopentane, and stored at –80°C. Muscles were longitudinally sectioned at

6 µm and fixed with 2% paraformaldehyde (PFA). Sections were permeabilized using 0.1% Tween 20 in phosphate-buffered saline (PBS), blocked with bovine serum albumin (BSA), and incubated overnight at 4°C with primary antibodies. Tissue sections were washed with PBS, then incubated for 1 hr at room temperature with Alexa Fluor conjugated secondary antibodies (Thermo Fisher) prior to mounting with ProLong Gold (Molecular Probes).

Sections with only α-actinin were imaged using Leitz DMRBE microscope with a Leica DCF480 digital camera. Sarcomere lengths were measured using the 1D line tool within ImageJ as the distance between peak fluorescence signals. For titin-epitope-specific immunohistochemistry, sections were permeabilized using 0.1% Tween 20 in PBS, blocked with 10% BSA 5% goat serum in permeabilization buffer, and incubated overnight at 4°C for each of the primary antibodies. Secondary antibodies were added for 1 hr at room temperature between primary antibodies. Sections with titin-epitope-specific antibody staining were imaged using a Leica TCS-SP8 confocal microscope equipped with HyVolution deconvolution software. Measurements were made using Leica LASX software.

For RBM20-AAV rescue experiments, tissue sections were permeabilized with 0.5% Triton for 10 min, washed 3 × 5 min with TBS, incubated with Image-iT FX Signal Enhancer (Thermo Fisher) for 30 min, and blocked for 1 hr at room temperature with 5% BSA, 5% normal goat serum, 1% glycine, 0.1% Triton X-100 in TBS. Following blocking, sections were incubated overnight at 4°C with TTN-PEVK (rabbit, Myomedix, 1:500) and myomesin (mouse, DSHB, 1:1000). The following day, sections were washed with TBS-Tween (0.1%) before incubation with secondary antibodies (goat anti-mouse Alexa Fluor 488 1:500, goat anti-rabbit Alexa Fluor Plus 405 1:500) in blocking buffer. Following post-secondary wash (2 × 5 TBS-T, 1 × 5 TBS), sections were post-blocked with goat anti-rabbit unconjugated fab fragment antibodies (1:200, Jackson ImmunoResearch), before incubation overnight with α-actinin antibody (Abcam, rabbit, 1:1000) in blocking buffer. The following day, sections were washed before incubation with secondary antibody (goat anti-rabbit Alexa Fluor 647). After secondary incubation, sections were washed (2 × 5 TBS-T, 1 × 5 TBS, 1 × 5 Ultrapure Water) before being mounted with ProLong Glass Antifade. Sections were imaged using a Nikon Spinning Disk SR microscope CSU-X1 using ×100 oil-objective lens, standard zoom. Z-stacks (0.3 µm steps) were obtained and reconstructed in ImageJ for analysis.

## Vertical agarose gel electrophoresis

TA muscles were pulverized under liquid nitrogen using a mortar and pestle to prevent fiber type and muscle region bias. Muscle powder was dissolved in sample buffer consisting of 8 M urea, 2 M thiourea, 0.05 M Tris base, 75 mM dithiothreitol (DTT), and 3% SDS at a 60:1 volume–muscle ratio to ensure titin solubilization (*Warren et al., 2003*). Protein concentrations were measured using the RC-DC protein assay (Bio-Rad) and stored at –80°C until use. Samples were resolved on a vertical agarose gel consisting of 1.0% w/v Seakem Gold Agarose (Lonza), 30% glycerol, 50 mM Tris base, 0.384 M glycine, 0.1% SDS run at 5 mA/gel for 30 min, then 7.5 mA/gel for 4 hr at 4°C (*Warren et al., 2003*). Titin protein was visualized using Sypro Ruby (Invitrogen) and imaged using a ChemiDoc MP System (Bio-Rad) at 450 nm. Bands were densitometrically analyzed using Bio-Rad ImageLab software as long T1 titin (upper), short T1 titin (lower), T2 titin (a major titin degradation product), and total titin (all bands combined).

## RNA isolation and RNA sequencing

RNA was isolated using procedures common to our lab for next-generation RNA sequencing (*Schroder et al., 2015*; *Hodge et al., 2015*; *Terry et al., 2018*). Briefly, muscle powder was homogenized in TRIzol (Invitrogen) according to the manufacturer's directions, then isolated using QIAGEN RNeasy Mini Kit. RNA samples were treated with TURBO DNase (Ambion) to remove genomic DNA. RNA integrity numbers (RIN) were determined using the Agilent 2100 Bioanalyzer. RNA samples with RIN values >8 were used for sequencing library preparation. Then, 250 ng of total RNA was used to prepare barcoded RNAseq libraries using Stranded RNA-Seq Kit with RiboErase (KAPA Biosystems). Samples were sequenced on the HiSeq 2500 (*Figure 2*) or NovaSeq 6000 (*Figure 6*; Illumina) using a 2 × 100 kit to a read depth >45 million reads/sample. Reads were checked using FastQC before mapped to mm10 with HISAT2 (*Kim et al., 2015*). Gene expression data were collected using HTSeq and DESeq2 analysis packages (*Love et al., 2014*), and splicing analysis was performed using the protocol outlined by *Schafer et al., 2015*. Exon usage data is reported as PSI and change in PSI (ΔPSI;

iMS*Bmal1*[-/-] - iMS*Bmal1*[+/+]). Data have been deposited in NCBI's Gene Expression Omnibus and are accessible using the GEO accession number GSE189865.

## Cell culture

Our generation of eGFP-labeled α-actinin-2 C2C12 cells has been previously described (*Denes et al., 2019*). Briefly, C2C12 myoblasts obtained from ATCC were stably transfected with ACTN2-pEGFP gene fusion using the PiggyBac Transposon system and selected using puromycin and flow cytometry. Myoblasts were maintained on plastic cell culture dishes in Dulbecco's Modified Eagle Medium (DMEM) supplemented with 10% fetal bovine serum and 1% penicillin-streptomycin in a humidified incubator kept at 37°C and 5% $CO_2$. When cells reached 70% confluency, they were separated and plated on micropatterned gelatin hydrogels. Gelatin hydrogels and PDMS stamps used for patterning were prepared as previously described (*Denes et al., 2019*; *Bettadapur et al., 2016*). Once 100% confluent, cells were serum-restricted with differentiation medium (DMEM, 2% horse serum, 1% penicillin-streptomycin) and left to differentiate for 10 days. Media were replenished each day, and all myoblasts were used at passage 8 or earlier. This cell line was validated using RNA deep sequencing (*Denes et al., 2019*) as well as through manufacturer certification and has tested mycoplasma negative.

## Titin-specific U7 snRNPs

Titin-specific, modified U7 snRNAs were generated using methods previously published to direct dystrophin splicing (*Brun et al., 2003*; *Echevarría et al., 2018*; *Goyenvalle and Davies, 2011*). The donor splice site on exon 51 and the acceptor splice site on exon 89 were targeted to induce the exon-skipping event studied through RNA sequencing of iMS*Bmal1*[-/-] and iMS*Bmal1*[+/+] muscle. Sequences were generated to block these splice sites according to published standards and cloned into pAAVsc-CB6-PI-mCherry (*Goyenvalle and Davies, 2011*). Briefly, a base U7smOpt vector was generated from the wildtype U7 snRNA gene using published primers and QuikChange II Site-Directed Mutagenesis Kit. Forward- and reverse-antisense primers were then used to insert titin-specific sequences into the pAAV-U7smOpt vector with a free tail harboring canonical binding sites for hnRNP A1/A2 to increase exon-skipping efficiency (*Goyenvalle and Davies, 2011*). Each cloned plasmid was confirmed by Sanger sequencing. Then, 150 ng of each plasmid were transfected with X-tremeGENE 9 (Roche) in C2C12 myoblasts immediately after trypsinization (*Escobedo and Koh, 2003*). Day 10 myotubes were fixed in 2% PFA for 4 min, then immunostained and imaged using the same conditions as with cryosectioned TA muscle; however, the antibody against the Z1Z2 region of titin was not used due to insufficient penetrance into myotubes.

## Quantitative real-time PCR (qRT-PCR)

Total RNA from iMS*Bmal1*[-/-] and iMS*Bmal1*[+/+] TA muscles was extracted using TRIzol Reagent (Ambion) and purified using the RNeasy Mini Kit (QIAGEN). Then, 500 ng total RNA was reverse-transcribed using SuperScript III first-strand cDNA synthesis system (Invitrogen). Primers were generated using previously published sequences for *Rbm20* and *Rpl26* (*Hodge et al., 2019*; *van den Hoogenhof et al., 2018*). qRT-PCR was performed on a QuantiStudio 3 thermal cycler (Applied Biosystems) from 10 ng of cDNA in a total volume of 20 μl containing Fast SYBR Green Master Mix (Applied Biosystems) and 400 nM of each primer. *Rbm20* was normalized to corresponding *Rpl26*, and relative quantification was calculated using the ΔΔCT method. Gene expression is shown as the relative fold change of expression in iMS*Bmal1*[-/-] muscle compared to iMS*Bmal1*[+/+] samples.

## Preparation of tibialis anterior muscle samples for LC-MS

Small ~1–2 mg pieces of TA muscle were obtained from five iMS*Bmal1*[+/+] and six iMS*Bmal1*[-/-] mice. The proteins were solubilized, reduced, alkylated, and digested to peptides using trypsin, as previously described (*O'Leary et al., 2019*). In brief, each piece of muscle was placed in a glass-bottom dissection chamber containing 150 μl 0.1% RapiGest SF Surfactant (Waters) and mechanically triturated with forceps. The solubilized proteins were reduced by addition of 0.75 μl 1 M DTT and heating (100°C, 10 min). Proteins were alkylated by addition of 22.5 μl of a 100 mM iodoacetamide (Acros Organics) in 50 mM ammonium bicarbonate (30 min, ~22°C, in the dark). The proteins were cleaved into tryptic peptides by addition of 25 μl of a 0.2 μg/μl trypsin (Promega) in 50 mM ammonium bicarbonate and

incubation (18 hr, 37°C). The samples were dried down by centrifugal evaporation and reconstituted in 100 μl of a 7% formic acid in 50 mM ammonium bicarbonate solution to inactivate trypsin and degrade RapiGest (1 hr, 37°C). The samples were dried down once more and reconstituted in 100 μl 0.1% trifluoroacetic acid (TFA) for further cleavage of RapiGest (1 hr, 37°C). The samples were dried down a final time, reconstituted in 150 μl 0.1 TFA, centrifuged for 5 min at 18,800 RCF (Thermo, Sorvall Legend Micro 21R), and 125 μl of the supernatant were removed for analysis by mass spectrometry.

## Liquid chromatography–mass spectrometry (LC-MS)

Tryptic peptides were separated via high-pressure liquid chromatography (LC) and analyzed by mass spectrometry (MS) as previously described (*O'Leary et al., 2019*). In brief, a 20 μl aliquot of each sample was injected into a Waters XSelect HSS T3 column using a Dionex UltiMate 3000 LC system. The effluent was directly infused into a Q Exactive Hybrid Quadrupole-Orbitrap mass spectrometer (Thermo) through an electrospray ionization source. Data were collected as Xcalibur .raw files in a data-dependent MS/MS mode, with the five most abundant ions selected for fragmentation.

## Identification of peptides and matching to titin exon position

Titin peptides were identified from the Xcalibur .raw files and their LC-MS peak areas were quantified using Thermo Proteome Discoverer 2.2 (PD) software as previously described (*O'Leary et al., 2019*). The protein database used to identify the peptides was generated by translation of the Ensembl ENSMUST00000099981.10 titin transcript. This nucleotide sequence contained 347 of the possible 363 genomic exons. However, the exon numbering in the Ensembl database referred to the exon position in the transcript. To number the exons according to the genomic sequence, the Ensembl ENSMUST00000099981.10 nucleotide sequence was compared to the BN001114.1 entry in GenBank and numbered accordingly. This genomic exon numbering scheme is commonly used in the current literature (*Guo et al., 2010*). The peptides identified within the LC-MS data were then labeled in accordance with the genomic exon(s) from which they were translated.

## Relative expression of peptides from exons 52–69, 70–79, and 80–88 in the iMS*Bmal1*$^{-/-}$ and iMS*Bmal1*$^{+/+}$ muscle

To determine whether there were differences in the expression of peptides from exons 52–69, 70–79, and 80–88, peptide abundances were compared by a pairwise analysis. Peptides with minimal LC-MS peak areas, defined as having a median abundance of less than 1.5E7 units within the control group, were excluded to ensure the use of robust signal intensities. Each remaining LC-MS peak area was normalized by dividing its value by the average LC-MS peak area of the peptides coming from exons 256–363 in the sample. This region was selected for normalization because there is a lack of evidence for alternative splicing (*Guo et al., 2010*). The median normalized abundance of each peptide within the iMS*Bmal1*$^{-/-}$ and iMS*Bmal1*$^{+/+}$ samples was determined. The average abundance and standard deviation of all of the peptides from each of the three exons were determined and plotted using ggplot2 in R.

## Western blot

Protein lysates were prepared by homogenizing approximately 20 μg of TA muscle from iMS*Bmal1*$^{+/+}$ and iMS*Bmal1*$^{-/-}$ mice in RIPA buffer containing protease inhibitors using a Bullet Blender (Next Advance). Then, 40 μg of total protein were loaded into each well for SDS-PAGE (10%). Proteins were transferred with a semidry apparatus (Bio-Rad) onto PVDF membrane, then blotted overnight at 4°C with anti-RBM20 antibody (Myomedix) or anti-γ-tubulin (Sigma) antibodies. Secondary antibodies were applied at room temperature for 1 hr, and the blot was visualized using a ChemiDoc MP System (Bio-Rad). RBM20 bands were quantified with Image Lab software and normalized to the γ-tubulin loading control.

## Targeted ChIP-PCR in adult skeletal muscle from C57Bl/6J mice

MYOD1, BMAL1, and CLOCK ChIP-seq data (accession numbers GSE122082 and GSE143334) recently published by our lab were searched for binding peaks in *Rbm20* using UCSC Genome Browser (*Gabriel et al., 2021*; *Hodge et al., 2019*; *Cao et al., 2010*). Following identification of overlapping peaks, putative regulatory regions identified by the ENCODE Registry of *cis*-Regulatory

Elements were scanned for canonical E-box sequences 5′-CANNTG-3′ (*Moore et al., 2020*). Primers flanking one of these E-box containing peaks in *Rbm20* were designed and used for ChIP-PCR to confirm binding.

For ChIP-PCR, approximately 800 mg of adult gastrocnemius muscle was homogenized in cell lysis buffer (10 mM HEPES, pH 7.5, 10 mM $MgCl_2$, 60 mM KCl, 300 mM sucrose, 0.1 mM EDTA, pH 8.0, 0.1% Triton X-100, 1 mM DTT, protease inhibitors) and prepared as previously described (*Hodge et al., 2019*). After pre-clearing with Protein A/G beads, supernatant was split into equal parts and incubated overnight at 4°C with MYOD1 antibody, CLOCK antibody, BMAL1 antibody, or pre-immune serum prior to DNA recovery with a PCR purification kit (QIAGEN). qRT-PCR was performed with *Rbm20* intron 1 primers. Data are presented as percent of input Ct, and significance was determined by comparing experimental ChIP values to those of the IgG control.

### Dual-luciferase assays

The promoter region of *Rbm20* (–500 to +138) was cloned into pGL3-basic from mouse genomic DNA to generate *Rbm20*-P-Luc. The putative enhancer element (389 bp) was then cloned into *Rbm20*-P-Luc upstream of the promoter region. All cloned fragments were confirmed via Sanger sequencing.

Luciferase assays were performed from lysates of C2C12 myotubes (*Andrews et al., 2010*). In each reaction, 50 ng luciferase reporter and 150 ng expression vectors were transfected into C2C12 myoblasts in 24-well plates (50,000 cells per well) using X-tremeGENE 9 (Roche). Also, 5 ng modified pRL null plasmid (*Hodge et al., 2019*) was used as a transfection control. C2C12 myoblasts were transfected as previously described (*Hodge et al., 2019*). Luciferase activity was measured using the Promega Dual Luciferase assay system. *Firefly* luciferase activity was normalized to *Renilla* luciferase activity. Normalized luciferase activity for each transfection is plotted as average fold change relative to the pGEM empty vector control. Results were analyzed using a two-way ANOVA.

### Generation of RBM20-AAV and RBM20 rescue

Mouse *Rbm20* cDNA was cloned into pAAV-CMV with a C-terminal HA-tag and confirmed by Sanger sequencing. AAV was produced using transient transfection of HEK 293T cells and CsCl sedimentation by the University of Massachusetts Medical School Viral Vector Core as previously described (*Sena-Esteves and Gao, 2020*). Vector preparations were determined by ddPCR, and purity was assessed by SDS-PAGE and silver staining (Invitrogen). Expression of RBM20-HA was confirmed in C2C12 myoblasts via Western blot.

iMS*Bmal1*[fl/fl] mice were treated with tamoxifen or vehicle as described above. Four weeks after treatment, mice were injected with AAV into the TA muscle. AAV was diluted in sterile PBS, then $1 \times 10^{12}$ GC of RBM20-AAV or $0.5 \times 10^{12}$ GC of GFP-AAV were injected evenly throughout the entire TA of anesthetized mice. Muscles were collected 4 weeks after AAV injection and flash-frozen in liquid nitrogen for protein analyses.

### Statistics

Statistical analyses were performed using GraphPad Prism and R. Student's *t*-tests were used to compare two groups with *F*-tests to compare variance between groups. One-way ANOVA with Tukey's post-hoc analysis was used to detect significant differences in experiments with three groups. Values are reported as mean ± SEM. Statistical differences in the average abundances of peptides in LC-MS data were determined from the individual values for each peptide. To account for differences in the intrinsic ionization efficiency of peptides, the normalized peptide abundance values were subjected to a natural log transformation to produce a normally distributed array of values. The peptide values within each region were compared by Student's *t*-test. To adjust for multiple comparisons, a Bonferroni correction ($0.05/2 = 0.025$) was used to determine the threshold p-value of 0.025 for the two proximal Ig regions compared. Figures were created using a combination of GraphPad Prism, ggplot2, and karyoploteR packages within R, and Adobe Illustrator. $*p<0.05$, $**p<0.01$, $***p<0.005$, $****p<0.0001$.

## Acknowledgements

We thank Dr. David Moraga and the University of Florida's Interdisciplinary Center for Biotechnology Research NextGen DNA Sequencing Core. We also thank Anisha Saripalli for help with immunohistochemistry experiments. This work was supported by National Institutes of Health

grants F31AR070625 to LAR, R01AR066082 to KAE, DP5OD017865 to ETW, R01HL157487 to MJP, and the University of Florida. DWH and SL were supported by Leducq Foundation funding (13CVD04).

## Additional information

### Funding

| Funder | Grant reference number | Author |
|---|---|---|
| NIH Office of the Director | DP5OD017865 | Eric T Wang |
| National Institute of Arthritis and Musculoskeletal and Skin Diseases | R01AR066082 | Karyn A Esser |
| National Heart Lung and Blood Institute | R01HL157487 | Michael J Previs |
| Fondation Leducq | 13CVD04 | David W Hammers Siegfried Labeit |
| National Institute of Arthritis and Musculoskeletal and Skin Diseases | F31AR070625 | Karyn A Esser |
| National Institute of Arthritis and Musculoskeletal and Skin Diseases | R01AR079220 | Karyn A Esser |
| Wu Tsai Human Performance Alliance | AGR00023600 | Karyn A Esser |

The funders had no role in study design, data collection and interpretation, or the decision to submit the work for publication.

### Author contributions

Lance A Riley, Conceptualization, Data curation, Formal analysis, Methodology, Writing – original draft, Project administration, Writing – review and editing; Xiping Zhang, Joseph M Mijares, David W Hammers, Ping Du, Methodology; Collin M Douglas, Investigation, Writing – review and editing; Christopher A Wolff, Data curation, Investigation; Neil B Wood, Methodology, Writing – review and editing; Hailey R Olafson, Data curation, Methodology; Siegfried Labeit, Eric T Wang, Resources, Methodology; Michael J Previs, Resources, Methodology, Writing – review and editing; Karyn A Esser, Conceptualization, Resources, Supervision, Funding acquisition, Project administration, Writing – review and editing

### Author ORCIDs

Lance A Riley ![ORCID] http://orcid.org/0000-0001-7863-364X
Collin M Douglas ![ORCID] http://orcid.org/0000-0002-7601-3508
David W Hammers ![ORCID] http://orcid.org/0000-0003-2129-4047
Christopher A Wolff ![ORCID] http://orcid.org/0000-0002-5129-5692
Eric T Wang ![ORCID] http://orcid.org/0000-0003-2655-5525
Karyn A Esser ![ORCID] http://orcid.org/0000-0002-5791-1441

### Ethics

All experiments were conducted in accordance with the National Institutes of Health Guide for the Care and Use of Laboratory Animals and approved and monitored by the University of Florida Institutional Animal Care and Use Committee Protocols (IACUC numbers: 201809136, IACUC202100000018).

### Decision letter and Author response

Decision letter https://doi.org/10.7554/eLife.76478.sa1
Author response https://doi.org/10.7554/eLife.76478.sa2

# Additional files

## Supplementary files
• Transparent reporting form

## Data availability

Sequencing data have been deposited in GEO under accession code: GSE189865.

The following dataset was generated:

| Author(s) | Year | Dataset title | Dataset URL | Database and Identifier |
|---|---|---|---|---|
| Riley LA, Esser KA | 2021 | The Skeletal Muscle Molecular Clock Regulates Sarcomere Length Through Titin Splicing | http://www.ncbi.nlm.nih.gov/geo/query/acc.cgi?acc=GSE189865 | NCBI Gene Expression Omnibus, GSE189865 |

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
