## [Editor Report]

Riley et al. provide a fundamental study that advances our understanding of how muscle biology is regulated by the circadian clock. The authors use compelling methodology to reveal a novel molecular mechanism for circadian regulation of the muscle giant protein titin via the splicing factor RBM20. The work will be of broad interest to muscle and circadian biologists, with implications for muscle-related disorders.

---

## [Decision Letter]

**Decision letter after peer review:**

Thank you for submitting your article "The Skeletal Muscle Circadian Clock Regulates Titin Splicing Through RBM20" for consideration by *eLife*. Your article has been reviewed by 3 peer reviewers, and the evaluation has been overseen by Reviewing Editor Benjamin Prosser and Anna Akhmanova as the Senior Editor. The reviewers have opted to remain anonymous.

Essential revisions:

The reviewing team is enthusiastic about the work and each noted the generally high novelty, significance, and quality of the work. As you will gather from the full reviews below, some additional data, which we hope can be obtained fairly readily and potentially through existing samples, is needed to support key conclusions.

1) To support a key conclusion (and title of the manuscript), it is important to confirm circadian regulation of titin splicing using an approach orthogonal to Bmal KO. This could be done, for example, by examining whether titin splicing differs in muscle samples collected during the day or at night, or after circadian disruption.

(2) The restriction of analysis to only the TA muscle was viewed as a significant limitation, and the quantification of sarcomere length dispersion was also viewed as limited. Analysis in additional muscle groups outside the TA, and a more robust interrogation of sarcomere length dispersion across experiments (for example following rescue studies in Figure 6), will significantly strengthen these important aspects of the manuscript.

We hope that these experiments can be performed in a timely manner and look forward to evaluating a revised manuscript.

*Reviewer #2 (Recommendations for the authors):*

The paper can be strengthened by:

(1) Studying whether titin isoform expression undergoes circadian rhythms, using muscle samples collected at different times of the day,

(2) By studying sarcomere length dispersion in muscles stretched to different lengths, and

(3) By discussing (or experimentally addressing) the biological consequence of upregulating more compliant titin isoforms in greater detail.

*Reviewer #3 (Recommendations for the authors):*

1. Provide justification for why using TA muscle, not other muscles.

2. At least two muscles should be used to test RBM20 expression, Ttn splicing and sarcomere length.

3. Sarcomere length data are not consistent which need be clarified.

4. The authors may consider rescuing Bmal1 gene in iMS*Bmal1^-/-^* TA muscle to determine RBM20 expression, titin isoform switching and sarcomere length which would provide direct evidence for the link between Bmal1 and ttn splicing.

5. The authors may also consider making an interrupted circadian rhythms mouse model in wild type mice to prove the authors' claim that circadian clock regulates Ttn splicing through RBM20.

6. The authors need provide the detailed method about how they measure the proximal Ig length. The a-actinin staining is fuzzy.

7. RBM20 is a large protein (about 140 KDa) which has low solubility and is relatively hard to extract from muscle tissue. RIPA buffer may not be strong enough to extract all RBM20 proteins. The author may consider using the protein buffer for titin protein extraction described in the method section or using the same protein samples as for titin detection for western blotting.

---

## [Author Response]

Essential revisions:The reviewing team is enthusiastic about the work and each noted the generally high novelty, significance, and quality of the work. As you will gather from the full reviews below, some additional data, which we hope can be obtained fairly readily and potentially through existing samples, is needed to support key conclusions.(1) To support a key conclusion (and title of the manuscript), it is important to confirm circadian regulation of titin splicing using an approach orthogonal to Bmal KO. This could be done, for example, by examining whether titin splicing differs in muscle samples collected during the day or at night, or after circadian disruption.

The reviewers raise an important conceptual point that we did not make clearly enough in the text. The impact of the circadian clock on cell physiology has been classically viewed through its role in regulating rhythmic gene expression and processes. However, it has become apparent that the core clock factors can modulate expression of non-rhythmic genes and that is the function we have identified in this study.

Another topic for clarification: We recognize that the titin protein has a half-life significantly longer then 24hrs. We also note that our lab and others report that *Rbm20* mRNA does not oscillate over time of day (Schroder et al., 2015 and Pizarro et al., 2013) so is not easily recognized as a circadian clock-controlled gene. Our work points toward a model in which the muscle clock contributes to maintenance Rbm20 expression and when there are chronic changes to clock function this leads to changes in Rbm20 expression with downstream effects on titin splicing. Accordingly, we have edited our manuscript in several places to clarify this point.

Lastly, we agree that having a different model is very important. Thus, we have generated new data from a model of circadian disruption using “chronic jet lag” to environmentally disrupt the muscle clock in wildtype mice and ask if this was sufficient to change *Rbm20* expression. Using this model, we found that *Rbm20* expression was disrupted in the quadriceps muscles of the jet lag group supporting our model linking the function and regulation of the muscle clock with *Rbm20* expression. This new data is now added to Figure 5. We also note that with our two time point collection we do see significant differences in Rbm20 mRNA expression in the quadriceps muscles of wildtype mice under normal conditions. This was unexpected as we do not see similar differences in Rbm20 mRNA levels in either the TA or Gastrocnemius muscles. However, this time of day difference is consistent with the muscle clock contributing to Rbm20 gene expression. We note these new findings in the Results section.

(2) The restriction of analysis to only the TA muscle was viewed as a significant limitation, and the quantification of sarcomere length dispersion was also viewed as limited. Analysis in additional muscle groups outside the TA, and a more robust interrogation of sarcomere length dispersion across experiments (for example following rescue studies in Figure 6), will significantly strengthen these important aspects of the manuscript.

The TA was chosen for analysis for several reasons, primarily that the predominant titin isoforms are small enough that subtle changes can be assessed using SDS-VAGE and that the pennation angle of this muscle is such that longitudinal sections of the entire muscle could be assessed. We do understand that this is a limitation of our analysis and have since interrogated this mechanism in the gastrocnemius muscle in our model with and without the rescue of *Rbm20* in iMS*Bmal1^-/-^* muscle and by including an analysis of the quadriceps muscle and TA in the “jet lag” model. We note that the focus of this work is on the identification of a very novel pathway for regulation of titin splicing in skeletal muscle. This pathway has never been identified or even speculated to date. We provide evidence for this pathway in both TA and Gastrocnemius muscle and feel that more analysis across different muscles, including heart, would be an important topic a future study.